# Phenolic Extract from Olive Leaves as a Promising Endotherapeutic Treatment against *Xylella fastidiosa* in Naturally Infected *Olea europaea* (var. *europaea*) Trees

**DOI:** 10.3390/biology12081141

**Published:** 2023-08-17

**Authors:** Veronica Vizzarri, Annamaria Ienco, Cinzia Benincasa, Enzo Perri, Nicoletta Pucci, Erica Cesari, Carmine Novellis, Pierluigi Rizzo, Massimiliano Pellegrino, Francesco Zaffina, Luca Lombardo

**Affiliations:** 1CREA Research Centre for Olive, Fruit and Citrus Crops, 87036 Rende, Italy; veronica.vizzarri@crea.gov.it (V.V.); annamaria.ienco@crea.gov.it (A.I.); cinzia.benincasa@crea.gov.it (C.B.); enzo.perri@crea.gov.it (E.P.); carminenovellis89@gmail.com (C.N.); rizzo.pierluigi@yahoo.com (P.R.); massimiliano.pellegrino@crea.gov.it (M.P.); francesco.zaffina@crea.gov.it (F.Z.); 2CREA Research Centre for Plant Protection and Certification, 00156 Rome, Italy; nicoletta.pucci@crea.gov.it (N.P.); erica.cesari@crea.gov.it (E.C.)

**Keywords:** *Olea europaea* L., *Xylella fastidiosa*, polar phenols, endotherapy

## Abstract

**Simple Summary:**

The currently applied containment strategies against the quarantine bacterium *Xylella fastidiosa*, the causative agent of olive quick decline syndrome, resulted to be only partially effective in limiting the further spread towards other uninfected areas. This study reports the promising results of an endotherapeutic trial conducted using a phenolic extract from olive leaves to counteract the bacterium, in comparison with a solution based on garlic powder and potassium phosphite, in naturally infected mature olive trees in Apulia Region. The two years in planta trial demonstrated a statistically significant effectiveness of polar phenols in stimulating the vegetative growth of the treated trees, likely due to the bacteriostatic effect shown in an in vitro test. A similar effect in limiting the planktonic growth of the bacterium was also highlighted for the solution based on garlic powder, which, however, did not seem to have significant effects in stimulating the vegetative growth of plants, unlike potassium phosphite which instead confirmed its plant growth booster action.

**Abstract:**

(1) Background: Since 2013, the pathogenic bacterium *Xylella fastidiosa* has been severely affecting olive production in Apulia, Italy, with consequences for the economy, local culture, landscape and biodiversity. The production of a phenolic extract from fresh olive leaves was employed for endotherapeutic injection into naturally infected olive trees by *Xylella fastidiosa* in Apulia region, Italy. (2) Methods: The effectiveness of the extract was tested in vitro and in planta in comparison with analogous treatments based on garlic powder and potassium phosphite. (3) Results: The uptake of phenolic compounds from olive leaves through a trunk injection system device resulted in a statistically significant increase in leaf area index and leaf area density, as well as in the growth of newly formed healthy shoots. Plant growth-promoting effects were also observed for potassium phosphite. Moreover, the bacteriostatic activities of the phenolic extract and of the garlic-powder-based solution have been demonstrated in in vitro tests. (4) Conclusions: The results obtained and the contained costs of extraction make the endotherapeutic treatment with phenolic compounds a promising strategy for controlling *X fastidiosa* to be tested on a larger scale, although the experiments conducted in this study proved not to be suitable for centenary trees.

## 1. Introduction

*Xylella fastidiosa* (Xf) [1] is a Gram-negative bacterium that colonizes the xylem vessels of more than 600 plant species, causing diseases such as citrus variegated chlorosis (CVC), Pierce’s disease (PD) of grapevine, and coffee leaf scorch (CLS) [2,3]. The main symptoms associated with Xf infection are marginal and/or apical leaf scorching, twig and branch dieback and often the disease progression results in plant death. Nevertheless, in many cases, infections by this bacterium may be asymptomatic [4].

Xf was first detected in the European Union in 2013, in Apulian olive plants [3], where it was identified as the etiological agent of the so-called ‘olive quick decline syndrome’ (OQDS) [5]. More specifically, molecular analyses based on multi-locus sequence typing (MLST) showed that OQDS is caused by the subspecies *pauca* Sequence Type-ST-53 (or “De Donno” strain) [5,6]. Phylogenetic analyses indicate that the ST53 strain is probably originally from Costa Rica and it was realistically introduced in the Mediterranean area through the importation of ornamental plants, likely occurring in 2008 [7,8].

The host plant colonization process and the mechanisms that lead to the development of OQDS symptoms are to date not fully elucidated [9]. The generally accepted cause of the rapid desiccation of olive trees is the occurrence of vascular occlusion, which leads to the blockage of xylem transport and water stress [2]. Cardinale et al. [10], using a specific fluorescence in situ hybridization (FISH) probe (KO 210) for Xf, quantified the level of infection and vessel occlusion in both petioles and branches of naturally infected and non-infected olive trees. All symptomatic petioles exhibited bacterial colonization, particularly in the larger innermost vessels, and in several cases the vessels appeared to be completely occluded by a biofilm consisting of bacterial cells and extracellular matrix. Notwithstanding, the clogged xylem vessel in the branches showed the presence of tyloses and gums/pectin gels and no bacteria. Accordingly, De Benedictis et al. [11] described occlusions caused by plant-produced tyloses and gums/pectin gels with limited presence of bacterial cell aggregates. This mechanism which is induced by biotic and abiotic stressors aims to limit the diffusion of pathogens and phenomena of air embolism and water loss [12]. This defensive function would explain the comparable level of occlusion found in plants with different degrees of tolerance to the bacterium. Gums (but not tyloses) and callose-like structures were observed in the vessels of artificially inoculated olive plants of the cultivar Cellina di Nardò (susceptible) and Leccino (tolerant) [9]. The same study highlighted the presence of degraded pit membranes (PMs) exclusively in the infected vessels of the cv Cellina di Nardò, suggesting that the integrity of these porous structures between contiguous cell walls can somehow contain the spread of the pathogen by limiting their passage through adjacent vessels. Similarly, transcriptomic analysis of a tolerant (Leccino) and a susceptible (Ogliarola salentina) olive cultivar revealed that most of the down-regulated genes in Xf-infected plants encoded proteins involved in cell wall integrity and remodeling in both varieties, demonstrating the central role of this structure [13]. Moreover, most of the 38 genes up-regulated only in Leccino encoded for leucin-rich repeat receptor-like kinases (LRR-RLKs), proteins mainly involved in stress responses, thus likely inducing a differential reaction to infection in comparison to Ogliarola salentina. Eventually, contrasting results were found by correlating xylem vessels’ diameter and bacterial infection [14,15], insofar as larger vessels are generally more susceptible to embolism as well as tylosis formation [12].

Xf is transmitted by several species of xylem sap-feeding insects belonging to the order of Hemiptera (families Aphrophoridae, Cercopidae, Cicadellidae, Cicadidae and Clastopteridae) [16]. The meadow spittlebug *Philaenus spumarius* (Hemiptera, Aphrophoridae) identified as the main responsible for Xf spreading in Salento, the Apulian territory most hardly hit by the outbreak of Xf [17]. Although several promising trials are currently underway, to date, no definitive control measure is available, even considering that, in Italy, the use of antibiotics in agriculture is not allowed (Ministerial Decree 10.08.1971). Thus, the main measures adopted by the EU Commission to contrast the spread of OQDS (Commission Implementing Decision (EU) 2015/789 of 18 May 2015) are focused on the use of insecticides to control the vector population, and on the eradication of infected plants. In particular, articles 4 and 7 of Regulation (EU) 2020/1201 provide for the eradication of infected olive trees and the mandatory felling/eradication of all the plants specified as sensitive to Xf (including the olive tree) within a radius of 50 m from the infected one. Consequently, it is difficult to quantify the exact number of olive trees infected or killed/cut down due to Xf *pauca* ST53 in Apulia, but according to an estimate of the principal Italian organization of agricultural entrepreneurs, the extent of the damage should have affected around 21 million trees—~1/4 of the Apulian oliviculture [18]—as also asserted in a recent parliamentary question (available at https://aic.camera.it/aic/scheda.html?numero=3-00259&ramo=C&leg=19, accessed on 10 July 2023). Forecasts based on the progress rate of the epidemic provide an economic impact likely ranging from 1.9 to 5.2 (the worst case scenario being no more production) billion euros over a 50-year period (and from 0.6 to 1.6 billion euros in the case of replacement with tolerant plants) [19] and a 34% (short-term) and 30% (long-term) reduction in the delivery of ecosystem services (ES) through the erosion of the olive ecosystem value (e.g., protection of soil structure and biodiversity, estimated to diminish by 28%), primary production and cultural heritage [20].

Therefore, the importance and the contingence to find more effective and sustainable alternatives than the countermeasures implemented so far appear extremely clear. In this sense, as plant phenolic compounds are widely known for their role in disease resistance and have been shown to possess antimicrobial properties [21,22,23], the focus of this work was to develop an extraction method of the polar phenolic component from olive leaves and to test the obtained extract against Xf in vitro and in naturally infected plants by endotherapy, in comparison with analogous treatments based on garlic powder and potassium phosphite.

## 2. Materials and Methods

The research work was articulated in a first phase providing the development of the extraction protocol for the phenolic compounds from olive leaves. The obtained phenolic extract was then characterized for its composition and concentration. The solution was subsequently tested for its effectiveness against Xf in vitro. The extract was administered to plants (1) as such, (2) acidified with 1% ascorbic acid to limit phenomena of phenolic oxidation and degradation and (3) with the addition of potassium phosphite (commercial product Ionifoss, Cifosrl, S. Giorgio di Piano, BO, Italy). Potassium phosphite has been shown to trigger the plant’s immune system by inducing higher expression of plant defense genes [24] and by stimulating the formation of phytoalexins and hypersensitivity reactions [25]. Moreover, it acts as a biostimulant, improving nutrient uptake [26]. At this stage, it was decided to include a further treatment made up of a solution containing finely micronized garlic (*Allium sativum* L.) powder (commercial product Aglio, Cerrussas, Uboldo, VA, Italy). Garlic has long been known to contain several biologically active compounds with stimulating (allithiamine), bactericidal and fungicidal (alline, allicin) properties [27,28], and also exhibiting a repellent action against some insects [29]. To standardize the experimental conditions, potassium phosphite was also added in this case.

Eventually, the treatments were tested for their administrability as endotherapeutic treatments and their effectiveness against Xf on naturally infected olive plants in open-field conditions. The uptake rate of the solution was evaluated through a trunk injection device.

### 2.1. Phenolic Extraction from Olive Leaves

Olive leaves were harvested in April from trees of the cv Coratina present in the CREA OFA experimental olive orchard located in Rende (Italy). The cv Coratina was chosen for its high phenolic, and, in particular, oleuropein content in leaves [30,31,32]. The leaves, immediately after harvesting, were cleaned and subjected to mechanical crushing in water by means of a blender (1.5 kg of leaves in 3 L of distilled water). To facilitate the phenolic extraction, the mixture was kept under shaking in an ultrasonic bath (Fisher Scientific, Milan, Italy) for 15 min. To obtain the mechanical separation of the liquid component from the solid one, the paste obtained was subjected to centrifugation at 10,000 rpm for 5 min. The supernatant (aqueous phase) was collected and subjected to a second centrifugation at 10,000 rpm for 5 min and then to filtration on ashless filter paper (Whatman International Ltd., Maidston, England) to better clean the solution. Aliquots of extract subsequently filtered through PTFE filters (0.45 µm, Millipore Merk, Darmstadt, Germany) were analyzed for single phenols by LC-MS/MS and for total polar phenols by HPLC.

#### 2.1.1. LC–MS/MS Analysis for Single Phenols

Measurements were performed using an API 4000 Q-Trap mass spectrometer (MSD Sciex Applied Biosystem, Foster City, CA, USA) in negative ion mode using multiple reaction monitoring (MRM). The LC–MS experimental conditions were as follows: ionspray voltage (IS) −4500 V; curtain gas 22 psi; temperature 350 °C; ion source gas (1) 35 psi; ion source gas (2) 45 psi; collision gas thickness (CAD) medium; entrance potential (EP), declustering potential (DP), entrance collision energy (CE) and exit collision energy (CXP) were optimized for each transition monitored. The chromatographic separation was achieved by means of an Eclipse XDB-C8-A HPLC column (5 µm particle size, 150 mm length and 4.6 mm i.d.; Agilent Technologies, Santa Clara, CA, USA) at a flow rate of 300 µL min^−1^ with an injection volume of 5 µL. A binary mobile phase made up of 0.1% aqueous formic acid (A) and methanol (B) was gradient programmed to increase B from 5% to 100% in 15 min, hold for 5 min and ramp down to original composition (95% A and 5% B) in five minutes. The total elution time was 25 min per injection.

#### 2.1.2. Quantitative Analysis for Single Phenols

Quantitative analyses were performed by external calibration curves built using a least-squares linear regression analysis. For this purpose, standard stock solutions of homovanillic acid (Omo), catechol (Cat), catechin (Cate), caffeicacid (Caf), vanillic acid (Van), o-cumaric acid (Ocum), p-cumaric acid (pCum), ferulic acid (Fer), apigenin (Ap), apigenin-7-O-glucoside (Ap7), diosmetin (Dios), hydroxytyrosol (Htyr), tyrosol (Tyr), oleuropein (Olp), luteolin (Lut), luteolin-7-O-glucoside (Lut7), luteolin-4-O-glucoside (Lut4), verbascoside (Verb) and rutin (Rut) were dissolved in methanol and further diluted with water/0.1% formic acid to obtain six calibration standards at concentrations in the range between 100 and 2000 µg mL^−1^. The correlation coefficients of the calibration curve ranged between 0.9995 and 0.9998. Each compound was monitored by the MRM mode whose scans, on the third quadrupole, were the main fragments of the deprotonated molecular ion [M-H]^−1^ produced in the first quadrupole. The standards were purchased from Extrasynthese (Genay Cedex, France) and Sigma–Aldrich (Riedel-de Haën, Laborchemikalien, Seelze, Germany). Methanol and formic acid were LC/MS grade and purchased from VWR International; aqueous solutions were prepared using ultrapure water, with a resistivity of 18.2MO cm, obtained from a Milli-Q plus system (Millipore, Bedford, MA, USA).

#### 2.1.3. HPLC for Total Polar Phenols Quantification

Measurements were performed using an HPLC-MS system (Agilent Technologies 1200 series liquid chromatography system). The analytes were separated on a C-18 RP column (5 μm particle size, 250 mm length × 4.6 mm i.d.) (Sigma–Aldrich, Riedel-de Haën, Laborchemikalien, Seelze, Germany) at a flow rate of 1 mL min^−1^ and an injection volume of 20 μL. The elution gradient followed the method described by the International Olive Council [33]. The total biophenol content (natural and oxidized oleuropein and ligstroside derivatives, lignans, flavonoids and phenolic acids), expressed in mg kg^−1^ of fresh leaves, was calculated by measuring the sum of the areas of the related chromatographic peaks according to the following formula:(mg kg^−1^) = [(ΣA) × 1000 × RRFsyr/tyr × (Wsyr)]/[Asyr × Wsyr](1)
where (ΣA) is the sum of the peak areas of the biophenols recorded at 280 nm; Asyr is the area of the syringic acid internal standard recorded at 280 nm; 1000 is the factor used to express the result in mg kg^−1^; W is the weight of the extract, in grams; RRFsyr/tyr is the multiplication coefficient for expressing the final results as tyrosol equivalents; Wsyr is the weight, in mg, of the syringic acid used as the internal standard in 1 mL of solution added to the sample.

### 2.2. Formulation of Treatments

Once the extraction was finalized, the following formulations were prepared to be tested: (1) 50 mL of pure phenolic extract (PE1); (2) 50 mL of phenolic extract acidified with 1% ascorbic acid (PE2); (3) 40 mL of phenolic extract acidified with 1% ascorbic acid with the addition of 10 mL of potassium phosphite (PE3); (4) garlic solution (G), consisting of 40 mL of sterile water in which 0.2 g finely micronized garlic powder dissolved by addition of 10 mL of potassium phosphite.

### 2.3. In Vitro Evaluation of Antibacterial Activity 

*Xylella fastidiosa* subsp. *fastidiosa* (Xff) strain Temecula1 (NCPPB 4605) was grown on PD2 agar medium for 7 days at 26 °C. The strain was maintained in the bacterial collection of the Research Centre for Plant Protection and Certification (CREA-DC), Rome, Italy, at −80 °C in phosphate-buffered saline (PBS) 1X containing 30% glycerol.

A stock solution of each formulation to test (G, PE1, PE2, PE3) was prepared by filtering with a 0.45 µm sterile filter and stored at −20 °C.

For the in vitro antibacterial assay, a broth macrodilution method was performed in glass tubes. Xff pure culture scraped off from PD2 plates was resuspended in a sterile glass tube containing 2.5 mL of PD2 liquid broth. After 5 days in a rotary shaking at 26 °C and 100 rpm, the adsorbance at 600 nm (OD_600_) of the bacterial suspension was spectrophotometrically measured with DeNovix Spectrophotometer DS-11 Fx + (Denovix Inc., Wilmington, DE, USA) and adjusted to a concentration of approximately 10^7^ colony forming unit (CFU) mL^−1^ (OD_600_ = 0.1) to be used as starter inoculum. Tubes containing 2.5 mL of PD2 broth (control tubes) or PD2 broth added with the substance at different concentrations (treated tubes) were inoculated with 100 μL of the starter bacterial suspension to reach a final concentration of 10^5^ CFU mL^−1^ (OD_600_ = 0.001). For each thesis, non-inoculated tubes (blank) were also used.

The in vitro antibacterial efficacy of each compound on the planktonic state of Xff was evaluated by reading the OD_600_ of 500 μL of each treatment after 0, 3 and 6 days post inoculation (dpi) by spectrophotometer. The biofilm production was also evaluated by the Crystal Violet assay and and determination of OD600 values (500 μL samples) according to Baldassarre et al. [34] was determined at 6 dpi.

For each inoculated tube, obtained data were normalized with absorbance resulting by blank tubes and expressed as a percentage. Statistical analysis was performed according to one-way ANOVA analysis (*p* < 0.05) and Dunnett’s multiple comparisons test (**** *p* < 0.0001, *** *p* < 0.001, ** *p* < 0.01, * *p* < 0.1).

The inhibition of the bacterial growth was evaluated through determination of the MIC (Minimum Inhibitory Concentration) value which is the lowest concentration of the compound able to inhibit bacterial growth. Each experiment was carried out twice.

### 2.4. In Planta Set up and Evaluation of the Xylematic Treatments

The on-field experimental trial was conducted from March 2021 to November 2022 in a private olive orchard of about one hectare in size; for the entire duration of the trial, the olive grove has undergone routine agronomic practices (normally implemented even before the start of the treatments): removal of excessively compromised branches (once a year, in February/March), copper treatments (twice a year, in April and October), fertilization (once a year, in March) and turf management (chemical weeding, twice a year in February/March and in August). The olive grove is located in the municipality of Lequile (Lecce, Apulia—40°18′12.89″ N, 18°5′53.02″ E, at 512 m of altitude), where Xf infection is endemic. This non-irrigated orchard includes olive trees grown in a polyconic vase form, planted between about 130 and 15 years ago. The endotherapeutic treatments were carried out on trees of the cv Cellina di Nardò, an autochthonous susceptible [9] olive cultivar. The selected plants showed clear signs of desiccation but were not yet seriously compromised. Six infected trees for each of the four treatments (PE1, PE2, PE3 and G) were chosen according to their -as homogeneous as possible- severity of symptoms, to be compared to the other 6 plants used as a control test (T). More specifically, 1 secular tree and 5 adult trees (~15–20 years old; that would have normally been in the full production phase) were surveyed per each treatment. As potassium phosphite turned out (see the Results and Discussions section) to be necessary for effective-treatment uptake, and as it is widely used in agriculture as a plant-growth booster, it was tested as a further treatment (KP) in five more plants, in order to evaluate the actual contribution of the phenolic extract in planta.

The plants’ stems were drilled (5.5 mm in diameter and 3 cm in depth to remain in the sapwood) to insert the Arbocap^®^ (Ital-Agro, Turin, Italy) syringes (50 mL) (Figure 1a). Arbocap^®^ is a refillable plastic capsule that works at a pressure of 0.5 bar, generated by an internal spring. At least three syringes were applied to each plant or to each principal branch diverging from the trunk (Figure 1b). Syringes were re-filled twice a month.

The effect of the treatments on the olive trees was evaluated through the indirect measurement of the canopy density before the start and at the end of the trial through a LAI-2200 PCA (plant canopy analyzer; Li-Cor, Inc., Lincoln, NE, USA). The LAI-2200 PCA is commonly used to estimate the total projected one-sided leaf area expressed per unit of ground surface area (leaf area index, LAI) or per unit of volume (leaf area density, LAD) by measuring diffuse light filtering through the plant canopy and the radiation above -outside- the canopy [35,36]. As this estimate takes into account all the light-blocking aerial plant structures (leaves, stems, branches, twigs), it seems particularly suitable to evaluate the progress of an infection causing the branches to dry up and the leaves to fall. Optical measurements for olive trees’ LAD and LAI were made in accordance with Lombardo et al. [37]. Briefly, data were collected along 24 points beneath the canopy of each tree at dawn, to limit the possible errors due to direct sunlight, in windless conditions and with the azimuthal field of view of the five concentric sensor rings limited by a cap of 45° towards the operator. Projected crown area (m^2^) and crown volume (m^3^) were measured according to Gómez et al. [38], with LAD calculated as the average of the measurements collected in the 24 points per tree taking into account the canopy height (= path length) of each point, while LAI was calculated as
(2)$$\mathrm{LAI}=\frac{LAD\times cV}{cA}$$
where cV and cA are canopy volume and area, respectively. The results were subjected to a parametric one-way analysis of variance (ANOVA), after having verified the normality and homoscedasticity of residuals, with a post hoc multiple-mean comparison (Tukey’s honestly significant difference, HS, test) at a 95% and 99% confidence level through PAST software v.4.10 [39].

Furthermore, the effect on vegetative growth was evaluated by the random selection of three healthy newly formed shoots per plant in the median region of the canopy. Shoot growth was recorded monthly in order to calculate the elongation rate of individual shoots. The shoots were identified and evaluated since the beginning of development (budburst), measuring from the shoot base insertion to the apical meristem.

The effect of treatment on shoot growth was evaluated by the Wilcoxon rank sum test using the “wilcox.test” function in the “dplyr” package [40]. Statistical analysis was conducted using R 3.4.1 [41].

## 3. Results and Discussions

### 3.1. Quantification of Total and Single Phenols in Olive Leaves

As expected, olive leaves were found to be a rich source of polar phenolic compounds. The concentration of total phenols in the aqueous leaf extract utilized in our experimental trial was 23,300 mg kg^−1^ with the analysis of individual phenols demonstrating that the secoiridoid oleuropein (OL) was the major component (mg kg^−1^, 45.6% of the total). The concentrations and compositions of phenolic extracts from olive leaves have been shown to vary according to extraction method, pedoclimatic conditions, orchard management system, sampling time and, of course, cultivar [42,43,44]. Accordingly, with particular regard to the cultivar Coratina, OL (and total phenol) concentration from this study was in the range obtained by Di Meo et al. [32] in an aqueous extract from leaves collected in March (oleuropein concentration raised up to 45 mg g^−1^ in leaves harvested in November) and by Difonzo et al. [45] from dry leaves through HPLC coupled to a diode array detector (DAD), while it was about a fifth of the values reported for frozen leaves macerated in 80:20 water:methanol [46] and two-fold higher than the result by Ghasemi et al. [31] from dried leaves in 50:50 water:methanol. However, regardless of the method, all these works converge in identifying OL as the most abundant compound of the polar phenolic component. As a consequence, the majority of the biological activity of the extract must probably be attributed to oleuropein. The well-known anti-inflammatory and antioxidant properties of OL have been attributed to its catecholic structure, able to scavenge the peroxyl radicals and break radical chain reactions resulting in more stable resonance structures [47,48]. The benefits of oleuropein and its constituents for human health have been demonstrated in several in vivo and in vitro studies (e.g., [49,50,51,52,53,54]), while in olive, OL derivatives have been shown to have a somewhat deterrent effect against *Bactrocera oleae* punctures [55,56]. Moreover, OL and its derivatives were demonstrated to inhibit the growth of bacterial pathogens such as *Listeria monocytogenes, Escherichia coli* O157:H7, and *Salmonella enteritidis* [57].

Beyond oleuropein, the aqueous leaf extract resulted to be very rich in substituted phenols such as catechol (2903 mg kg^−1^), hydroxytyrosol (662.71 mg kg^−1^), tyrosol (678.89 mg kg^−1^), verbascoside (39.11 mg kg^−1^) and, in smaller but appreciable quantities, cumaric (71.52 mg kg^−1^), ferulic (41.91 mg kg^−1^), caffeic (8.38 mg kg^−1^), vanillic (2.24 mg kg^−1^) and homovanillic (0.87 mg kg^−1^) acids, characterized by high antioxidant properties [58,59]. Many antioxidant, anti-inflammatory, anti-thrombotic, cytoprotective, vasoprotective and anti-microbial activities have been, also, associated with the class of compounds known as flavonoids. In this sense, rutin, known for its high level of free radical scavenging capacity [60,61,62], was found in our extract at 64.26 mg kg^−1^, while luteolin, luteolin-7-O-glucoside and luteolin-4-O-glucoside concentrations were 2023 mg kg^−1^, 1430 mg kg^−1^ and 704.04 mg kg^−1^, respectively. These last three compounds, together with rutin, have been found to possess important anti-microbial and anti-bacterial activities [63,64,65,66]. Furthermore, apigenin (7.26 mg kg^−1^), apigenin-7-O-glucoside (6.71 mg kg^−1^), diosmetin (569.94 mg kg^−1^) and catechin (480.90 mg kg^−1^) were also present in the aqueous extract. The obtained data are summarized in Table 1.

Ultimately, the extraction protocol employed in this trial was found to be time- and money-saving and to guarantee an adequate amount of total phenols for the subsequent experimental tests.

### 3.2. In Vitro Screening

Due to difficulties linked to the slow and discontinuous growth of Xf *pauca* experienced during the assay, preliminary in vitro tests were performed using Xf *fastidiosa* (Xff)treatment.

We evaluated both the ability of the substances to limit the planktonic growth of Xff after 3 and 6 dpi, and the ability to inhibit the biofilm formation 6 dpi. As reported in Figure 2, all the formulations tested (G, PE1, PE2, PE3) showed antibacterial efficacy on both the planktonic (Figure 2a) and biofilm states (Figure 2b) at the two concentrations 1:10 and 1:100, with statistically significant differences compared to control (Ctr). For the formulation G, only the concentration 1:10 of the substance showed an inhibition on Xff and statistically significant differences compared to control. MIC values for the substances were thus 1:10 for formulation G and 1:100 for formulations PE1, PE2, and PE3.

Xf is characterized by a fine balance between planktonic/biofilm growth regulated by a mechanism-defined quorum sensing (QS). When the balance is moved towards the biofilm state, there is an increased secretion of diffusible signal factors (DSFs) and cell-to-cell adhesion enhances.

The planktonic state allows Xf to move upstream and downstream along the xylem, thanks to the presence of vessel-degrading enzymes, and long and short pili. Biofilm state, tyloses and gum formation block the sap flow, causing water stress and nutritional deficiencies, leaf scorch, wilting and ultimately the death of infected plants. The biofilm phase is fundamental for bacterial transmission mediated by xylem-feeding hemipteran insect families and plays an important role in resistance against antimicrobial compounds [67].

Considering the 1:10 dilution, PE2 and PE3 induced the lowest planktonic growth after 6 dpi, although no statistically significant differences were found (Figure 2c). An explanation could be found in the presence of ascorbic acid which could have favored the maintenance over time of the biological activity of the phenolic component. On the other hand, potassium phosphite did not seem to have a particular role in the bacteriostatic action of the tested solutions, as there are practically no differences between the PE3 and PE2 treatment (with and without potassium phosphite, respectively).

These findings confirm the results of previous in vitro studies reporting the efficacy of several phenolic compounds in counteracting the growth of Xf [68,69]. In particular, among polyphenols, oleuropein showed the strongest bacteriostatic effect followed by luteolin-7-O-glucoside and verbascoside [69]. However, the inhibitory mechanism of phenolic compounds on Xf subspecies are still unknown.

Moreover, in a wider “antimicrobial sense”, olive leaf extracts were described to possess an activity against *Campylobacter jejuni*, *Campylobacter coli*, *Escherichia coli*, *Helicobacter pylori*, *Staphylococcus aureus*, *Salmonella enteritidis*, *Salmonella typhimurium*, *Listeria innocua* and *Bacillus cereus* [44,70,71,72,73]. Interestingly, according to Lee and Lee [71] leaf extracts were more effective than single phenols, suggesting a synergistic effect of the phenolic component.

### 3.3. In Planta Evaluation of the Endotherapeutic Treatments

The first result was that the endotherapeutic treatment as conceived in this trial was not suitable for secular trees. In fact, only very limited volumes of the solutions were absorbed by the plants. This could be due to the irreversible plugging of the xylem vessels after Xf infection, or to a greater inclination to vessel occlusion with the aging of the xylem because of cavitation and gum/tylosis formation [12,74]. Further studies are needed to clarify this crucial issue, as monumental olive trees constituting a worldwide recognized cultural and landscape heritage of Apulia have been particularly hit by the Xf outbreak.

A second empirical finding was that the phenolic extracts without the addition of potassium phosphite (PE1 and PE2) were not actively absorbed by plants, suggesting the need for a carrier for an efficient uptake of external fluids presenting different compositions compared to the sap. Even in this case, the limited volumes absorbed did not allow to evaluate, in a comparable way, the effectiveness of these two treatments in the open field. Therefore, in addition to the treatments with G and PE3, the effects of a solution consisting only of potassium phosphite were also studied, to evaluate the contribution of this biostimulant to the eventual growth of the trees. This treatment (KP), for G and PE3, was tested on five adult infected olive trees.

The effect of the treatments was evaluated with an LAI and LAD increase—if any—at the end of the trial. Leaf area index at the start of the trial ranged from 0.77 to 1.35 (average 1.09) m^2^ m^−2^, while leaf area density varied from 0.68 to 1.12 (average 0.88) m^2^ m^−3^, with average values per treatment reported in Table 2. These values fall in the lower range reported in Mezghani et al. [75] and Gómez et al. [38]. The homogeneity of the chosen plants was confirmed by the mean values of the five trees for each treatment (Table 2).

PE3 proved to be the most effective treatment, with statistically significant higher increases for both LAI and LAD in comparison to control (T) and the other two treatments (Figure 3). KP values resulted significantly higher than those of the untreated trees, while no relevant differences were found in comparison with G, which, in turn, presented significant differences with T only for LAD values and only at a 95% confidence level. The behavior of the trees treated with potassium phosphite alone can be traced back to its booster effect on plant growth, even if antibacterial effects on populations of *Candidatus liberibacter* [76] and *Streptomyces scabies* [77] have been described. The percentage increase varied from 20.19 (PE3), to 10.77 (KP), to 6.72 (G), to 2.46% (T) for LAI and from 9.25 (PE3), to 8.29 (KP), to 6.51 (G), to 0.26% (T) for LAD. It should be considered that, as previously specified, the plants were subjected to pruning of the branches that were definitively compromised and this may have, in some way, affected the measurement of some values. However, this lower percentage increase in LAI and LAD in untreated plants can be probably attributed to an increased progression of OQDS symptoms, either due to reduced vegetative growth of the shoots or to the need for a relatively more heavy pruning (or both), especially for a susceptible cultivar like Cellina di Nardò. Moreover, the differences were visually appreciable and consistent with the results obtained post treatment (Figure 4b,d,f) in comparison with the pretreatment (Figure 4a,c,e).

The effect of the treatment on the elongation rate of three healthy newly formed shoots per plant is shown in Figure 5. Growth was affected by treatment, with PE3 having the highest growth rate compared to the other two solutions, and presenting statistically significant higher values compared to T (*p*-value _PE3 vs. T_ = 0.0083; *p*-value _G vs. T_ = 0.9955; *p*-value _KP vs. T_ = 0.1238). This was probably due to the observation that the phenolic extract of olive leaves had a strong antibacterial effect on the whole plant, while the results of the KP treatment could be largely explained by the plant growth-promoting effect of potassium phosphite. Among the three tested treatments, G was found to be the least involved in the growth of the shoots.

Based on the results obtained, the endotherapeutic administration of phenolic extracts from olive leaves has proved to be an effective method for the treatment of Xf. Accordingly, endotherapy was used to evaluate the efficacy of Dentamet^®^, an internationally patented biofertilizer consisting of zinc–copper–citric acid biocomplex, resulting in the reduced multiplication rate of Xf in olive trees [78,79,80].

Furthermore, this approach would reduce the need to replace local varieties with less Xf-susceptible ones (like the cv Leccino and FS17 [81]), preserving the Apulian olive biodiversity, and, consequently, the Italian germplasm which, although including over 800 accessions, is mostly made up of minor varieties with limited territorial diffusion, also due to the orographic characteristics of the olive groves, often located in marginal areas [82,83].

## 4. Conclusions

Since 2013, the pathogenic bacterium X. fastidiosa has severely affected olive farming, local culture and biodiversity in Apulia. The need for alternative containment strategies are now indispensable to limit further spread towards other uninfected areas. In this sense, endotherapy is an efficient way to deliver agrochemicals, growth regulators, defense activators, plant biostimulant and fertilizers in many tree species, and it offers some advantages compared to other systems (i.e., foliar applications). This application method allows one to give bioactive compounds in an environmental friendly way to control bacteria, fungi and insects. This two-year study reports the promising results of an endotherapeutic trial conducted using an aqueous phenolic extract from olive leaves to counteract the bacterium in naturally infected olive trees in an Apulian locality where the disease is endemic, in comparison with a solution based on garlic powder and potassium phosphite. The in planta evaluation of the treatments showed a statistically significant greater efficacy of the phenolic solution in stimulating the trees’ vegetative growth expressed as a percentage increase in LAI and LAD, as well as on growth of newly formed healthy shoots. This appears to be probably due to the bacteriostatic effect as observed in the in vitro test. The garlic-powder-based solution also showed good action against the planktonic growth of the bacterium in vitro, but in the open-field experiment, it showed little effect on the foliage density, unlike potassium phosphite which confirmed its action as a plant-growth booster.

The contained costs and the efficacy tested both in vitro and in planta incentivize further studies providing the employment of this treatment on a larger scale. Moreover, further evidence from this trial is that the formulation of the solutions to be used in endotherapy is a crucial step in open-field experimentation. Eventually, the endotherapeutic treatment as reported in this study proved to be an unsuitable technique for centenarian plants.

## Figures and Tables

**Figure 1 biology-12-01141-f001:**
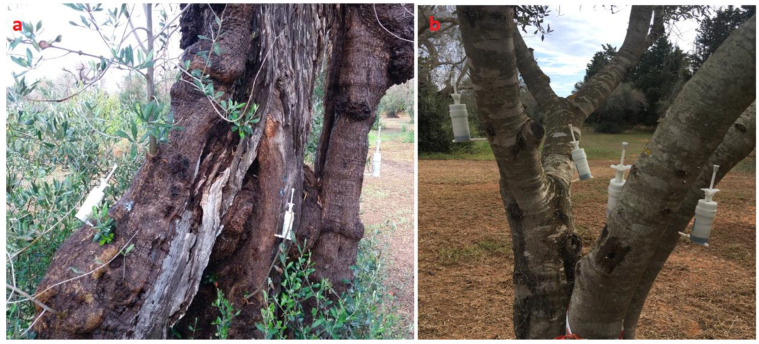
Positioning of the Arbocap^®^ syringes for endotherapeutic treatment in a secular plant (**a**) and in each of the 4 principal branches of an adult olive tree (**b**).

**Figure 2 biology-12-01141-f002:**
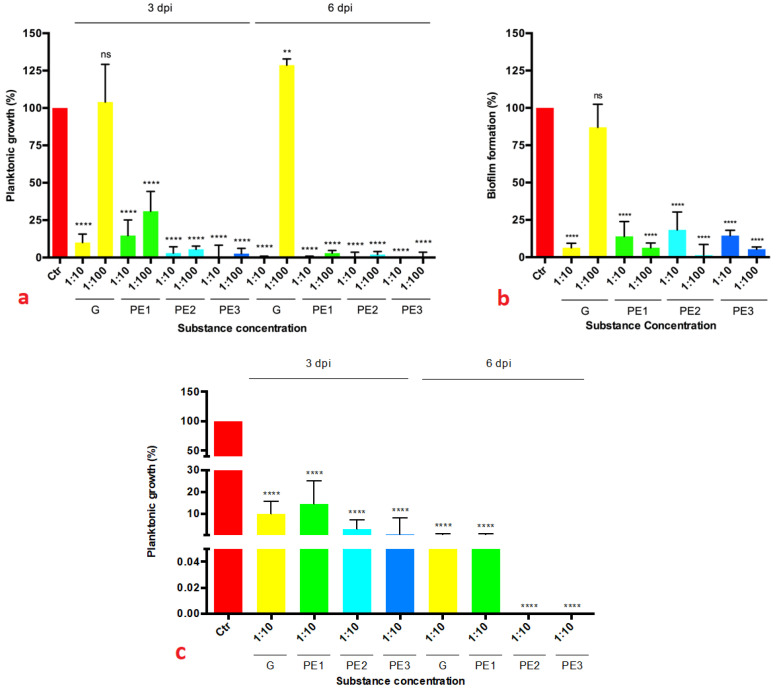
Percentage of planktonic growth (**a**) after 3 and 6 dpi (days post inoculation) and biofilm formation (**b**) after 6 dpi of *Xylella fastidiosa* subsp. *fastidiosa* (Xff) strain Temecula1 (NCPPB 4605) grown in PD2 broth added with four formulations (G, PE1, PE2, PE3) at different concentrations (1:10, 1:100) and PD2 without substances (Ctr). Values are means ± SD of three independent biological replicates, normalized with absorbance obtained by blank values (non-inoculation), of the most representative experiment of two independent experiments. (statistically significant difference obtained between control and the concentrations of the different formulations, according to one-way ANOVA analysis, Dunnett’s multiple comparisons test: **** *p* < 0.0001, ** *p* < 0.01).

**Figure 3 biology-12-01141-f003:**
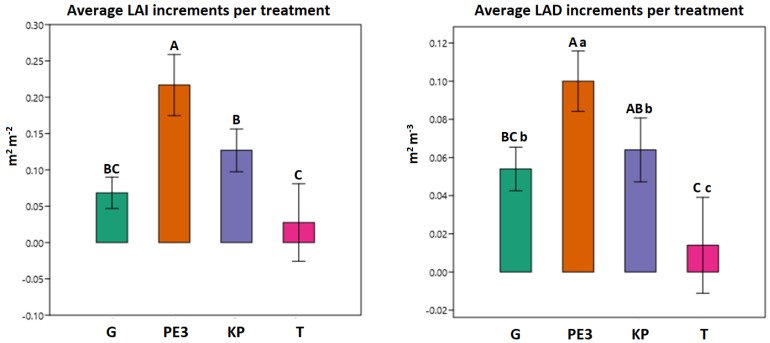
Effect of treatments on canopy density expressed as increments of leaf area index (LAI) and leaf area density (LAD). Different uppercase and lowercase letters indicate statistical significances at 99% and 95% levels, respectively, through post hoc multiple-mean comparison by Tukey’s honestly significant difference (HSD) test.

**Figure 4 biology-12-01141-f004:**
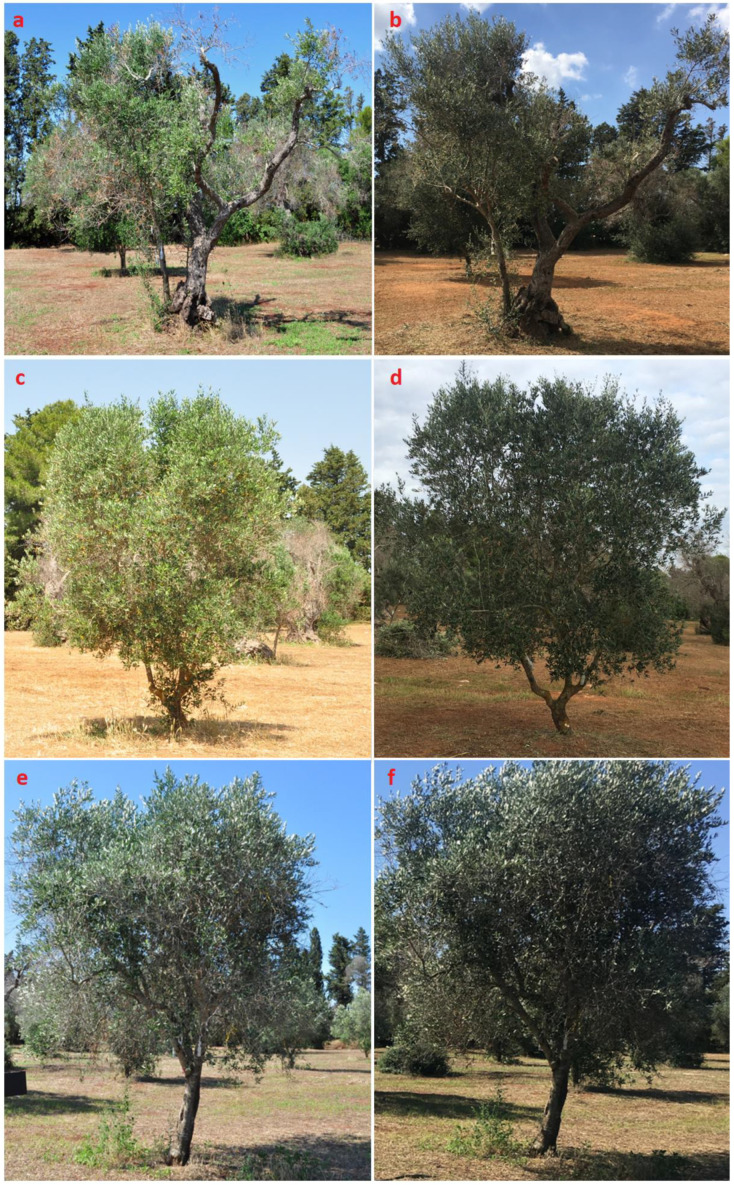
Pictures of three plants subjected to the three different treatments at the beginning (**a**,**c**,**e**) and at the end (**b**,**d**,**f**) of the experimental test. a and b: G; b and c: KP; E and f: EP3.

**Figure 5 biology-12-01141-f005:**
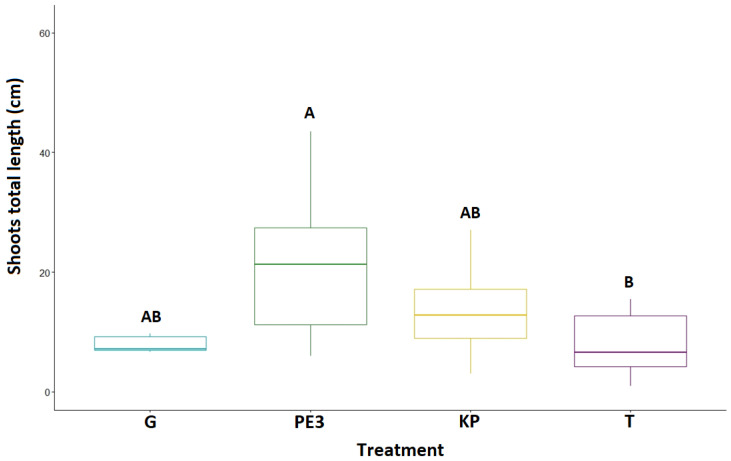
Box plots showing the elongation rate of shoots for the four treatments (G, PE3, KP and T). Different uppercase letters indicate statistical significances at 99% level.

**Table 1 biology-12-01141-t001:** Concentrations of single and total phenols in olive leaf water extract. The data are expressed in mg kg^−1^ of fresh leaves, and represent the mean values of tree replications with their relative standard deviations.

Substituted Phenols and Flavonoids	mg kg^−1^
Oleuropein	10,621 ± 121
Catechol	2903 ± 61
Hydroxytyrosol	662.71 ± 18.49
Tyrosol	678.89 ± 15.11
Diosmetin	569.94 ± 12.87
Catechin	480.90 ± 13.51
o-Cumaric acid	67.05 ± 10.12
Ferulic acid	41.91 ± 8.33
Verbascoside	39.11 ± 5.21
Caffeico acid	8.38 ± 4.02
p-Cumaric acid	4.47 ± 0.45
Vanillic acid	2.24 ± 0.85
Homovanillic acid	0.87 ± 0.04
Luteolin	2023 ± 31
Luteolin-4-O-glucoside	704.04 ± 25.41
Luteolin-7-O-glucoside	1430 ± 13.47
Rutin	64.26 ± 9.11
Apigenin	7.26 ± 1.03
Apigenin-7-O-glucoside	6.71 ± 0.96
Total polar phenols	23,300 ± 115

**Table 2 biology-12-01141-t002:** Average LAI and LAD values (±standard deviation; n = 5) per treatment at the start of the trial.

Treatment	LAI (m^2^ m^−2^)	LAD (m^2^ m^−3^)
G	1.02 ± 0.18	0.83 ± 0.09
PE3	1.07 ± 0.34	0.93 ± 0.22
KP	1.18 ± 0.11	0.90 ± 0.06
T	1.12 ± 0.21	0.88 ± 0.03

## Data Availability

The authors confirm that the data supporting the findings of this study are available within the article. Raw data are available from the authors (V.V., L.L., C.B, N.P. and E.C.) on request.

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
