# Peer review of "Phenolic Extract from Olive Leaves as a Promising Endotherapeutic Treatment against Xylella fastidiosa in Naturally Infected Olea europaea (var. europaea) Trees"

_biology, 2023, doi:10.3390/biology12081141_

Round 1

Reviewer 1 Report

I recomand that the manuscript biology-2532544 to be considered for publication as is.

Author Response

We want to thank the reviewer for his/her opinion

Reviewer 2 Report

This is the second time I am reviewing this article. The Authors accepted most of the suggestions and argued on some comments. Their opinion is respectable. However, I added (green italics) my further comments to some of their answers below. I am also attaching the text with some minor editings and further comments and suggestions.

Overall the article is interesting and can be accepted in the present form with some minor revisions.  I  am again wondering if journals such as  "Agronomy" and "Agriculture" are more suitable for this article provided that results of in field experiments are prevalently based on symptom severity on trees and no in planta biological test was performed.   

Comments to Author's response to the first revision round

The Authors tested the effectiveness of an extract from olive leaves in controlling Xylella fastidiosa infections. The trunk injection method they used is not original and is known by several decades.

-We have never argued otherwise. The originality lies in the tested compounds.

However I would have expected the Authors to cite other articles where the method or similar methods of tree trunk injection have been used for therapeutic purposes in order to support their choise; e.g.

Garbelotto et al. (2007) Phosphite Injections and Bark Application of Phosphite + Pentrabark™ Control Sudden Oak Death in Coast Live Oak. Arboriculture & Urban Forestry 33(5), 309–317.

Horner I. June 2020. Phosphite treatment by trunk injection in kauri. A Plant & Food Research report prepared for: Ministry for Primary Industries. Milestone No. 84468. Contract No. 37705. Job code: P/345168/01. PFR SPTS No. 19579.

Overall the article is of some practical interest but nothing to do with ahtioxidants which is the focus of this journal. I suggest the Authors to propose this article to journal like Agronomy or Agriculture (published by MDPI) after carefull revision.

A serious methodological concern is that the non- treated control is missing.

-The non-treated control is not missing at all (nor in the in vitro nor in the in planta trial), we wonder where the reviewer got this conclusion from.

I do apologize with the Authors for this misunderstanding. It was my fault. I was referring to in field tests where actually the Authors included a non –treated control. However it was my intention to point out that the control plants did not undergo the same treatment as the others, because during the experiment the intensity of pruning was directly proportional to the severity of symptoms therefore final results may have been influenced by the difference in ancillary routine treatments as stressed by the Authors themselves (Lanes 437-440). See next comment.

Please clarify this point.

Moreover symptomatic trees were pruned during the experiment and symptomatic branched were cut. Consequently, the most severe the disease was the most severe the pruning was: experimental conditions were not uniform and the results of treatments were altered.

-It was clearly stated in the text that the plants were chosen according to the uniformity of the symptoms of the disease, furthermore, the range of LAI and LAD at the start of the trial was very similar for all the treatments and the control, once again suggesting a certain uniformity of the trees. The higher need for pruning the control plants simply confirmed the effectiveness of the treatments employed

You cannot deny that by removing the withered twigs and branches you altered the values of both LAD and LAI. As far as the uniformity of trees before the experiment is concerned, Fig. 3 does not confirm this statement (in particular, I am referring to tree in Fig. 3a).

Despite these observations my opinion is your results deserve to be taken into serious consideration.

Reviewer 2

The Authors tested the effectiveness of an extract from olive leaves in controlling Xylella fastidiosa infections. The trunk injection method they used is not original and is known by several decades.

-We have never argued otherwise. The originality lies in the tested compounds.

However I would have expected the Authors to cite other articles where the method or similar methods of tree trunk injection have been used for therapeutic purposes in order to support their choise; e.g.

Garbelotto et al. (2007) Phosphite Injections and Bark Application of Phosphite + Pentrabark™ Control Sudden Oak Death in Coast Live Oak. Arboriculture & Urban Forestry 33(5), 309–317.

Horner I. June 2020. Phosphite treatment by trunk injection in kauri. A Plant & Food Research report prepared for: Ministry for Primary Industries. Milestone No. 84468. Contract No. 37705. Job code: P/345168/01. PFR SPTS No. 19579.

Overall the article is of some practical interest but nothing to do with ahtioxidants which is the focus of this journal. I suggest the Authors to propose this article to journal like Agronomy or Agriculture (published by MDPI) after carefull revision.

A serious methodological concern is that the non- treated control is missing.

-The non-treated control is not missing at all (nor in the in vitro nor in the in planta trial), we wonder where the reviewer got this conclusion from.

I do apologize with the Authors for this misunderstanding. It was my fault. I was referring to in field tests where actually the Authors included a non –treated control. However it was my intention to point out that the control plants did not undergo the same treatment as the others, because during the experiment the intensity of pruning was directly proportional to the severity of symptoms therefore final results may have been influenced by the difference in ancillary routine treatments as stressed by the Authors themselves (Lanes 437-440). See next comment.

Please clarify this point.

Moreover symptomatic trees were pruned during the experiment and symptomatic branched were cut. Consequently, the most severe the disease was the most severe the pruning was: experimental conditions were not uniform and the results of treatments were altered.

-It was clearly stated in the text that the plants were chosen according to the uniformity of the symptoms of the disease, furthermore, the range of LAI and LAD at the start of the trial was very similar for all the treatments and the control, once again suggesting a certain uniformity of the trees. The higher need for pruning the control plants simply confirmed the effectiveness of the treatments employed

You cannot deny that by removing the withered twigs and branches you altered the values of both LAD and LAI. As far as the uniformity of trees before the experiment is concerned, Fig. 3 does not confirm this statement (in particular, I am referring to tree in Fig. 3a).

Despite these observations my opinion is your results deserve to be taken into serious. Consideration.

Author Response

We thank the reviewer for the opinions and the suggestions expressed and the corrections made. Please see the attached file for the answers.

Reviewer 3 Report

The manuscript titled „Phenolic extract from olive leaves as a promising endotherapic treatment against Xylella fastidiosa in naturally infected Olea europaea (var europaea) trees” is about effects of endotherapic inoculation in naturally infected olive trees by Xylella fastidiosa in Apulia region.

For me, aim of this research is clean now, when I read the abstact.

Could you provide more detailed information about the experimental orchard (year / season of planting, orchard system especially canopy, irrigation, condition of the plants)? These information are still missing. Please add one sentence or two sentences about these information to undersand better the results and to be repeatable the trial-

It is also clean, when the leaves were collceted. The authors revised this information in the revised version of the paper.

In the main text the too long sentences were revised and replaced them to shorter versions.

English of the manuscript is corect, easy to follow and understand. 

Author Response

The manuscript titled „Phenolic extract from olive leaves as a promising endotherapic treatment against Xylella fastidiosa in naturally infected Olea europaea (var europaea) trees” is about effects of endotherapic inoculation in naturally infected olive trees by Xylella fastidiosa in Apulia region.

-For me, aim of this research is clean now, when I read the abstact.

-It is also clean, when the leaves were collceted. The authors revised this information in the revised version of the paper.

-In the main text the too long sentences were revised and replaced them to shorter versions.

-English of the manuscript is corect, easy to follow and understand.

We thank the reviewer for these comments.

Could you provide more detailed information about the experimental orchard (year / season of planting, orchard system especially canopy, irrigation, condition of the plants)? These information are still missing. Please add one sentence or two sentences about these information to undersand better the results and to be repeatable the trial-

In the text it is specified that:

The olive grove is located in the municipality of Lequile (Lecce, Apulia - 40°18'12.89"N, 18° 5'53.02"E, at 512 m of altitude -), where Xf infection is endemic. This non-irrigated orchard includes olive trees grown in a polyconic vase form, planted between about 130 and 15 years ago. […] The selected plants showed clear signs of desiccation but were not yet seriously compromised. […] More specifically, 1 secular tree and 5 adult trees (~15-20 years old; that would have normally been in the full production phase) were surveyed per each treatment.”

Furthermore, we have added that:

“… the olive grove has undergone routine agronomic practices (normally implemented even before the start of the treatments): removal of excessively compromised branches (once a year, in February/March), copper treatments (twice a year, in April and October), fertilization (once a year, in March) and turf management (chemical weeding, twice a year in February/March and in August).